# Pregnancy Complications in Pandemics: Is Pregnancy-Related Anxiety a Possible Physiological Risk Factor?

**DOI:** 10.3390/ijerph191912119

**Published:** 2022-09-25

**Authors:** Eman Abahussain, Manal Al-Otaibi, Khulud Al-Humaidi, Sultanah Al-Mutairi, Alexandra Al-Khatir, Amani Abualnaja, Sarah Al-Mazidi

**Affiliations:** 1Department of Neuroscience, Faculty of Medicine, Imam Mohammad Ibn Saud Islamic University, P.O. Box 5701, Riyadh 11432, Saudi Arabia; 2Faculty of Medicine, Imam Mohammad Ibn Saud Islamic University, P.O. Box 5701, Riyadh 11432, Saudi Arabia; 3Department of Obstetric and Gynecology, Faculty of Medicine, Imam Mohammad Ibn Saud Islamic University, P.O. Box 5701, Riyadh 11432, Saudi Arabia; 4Department of Physiology, Faculty of Medicine, Imam Mohammad Ibn Saud Islamic University, P.O. Box 5701, Riyadh 11432, Saudi Arabia

**Keywords:** mental health, pregnancy, anxiety, COVID-19, women’s health

## Abstract

Background: Birth and pregnancy complications increased by 10.2% during the 2019 coronavirus (COVID-19) pandemic. Pregnant women are at high risk for anxiety, which might trigger physio-logical stress, leading to pregnancy complications. Aim: This study aimed to investigate factors leading to antenatal anxiety during the COVID-19 pandemic. We also aimed to discuss our find-ings with regard to the current literature about pregnancy complications. Methods: This cross-sectional study interviewed 377 pregnant women and assessed anxiety using a validated 7-item general anxiety disorder (GAD-7) scale. Anxiety was related to physiological and demo-graphic parameters. Anxiety was subdivided into pandemic- and pregnancy-related anxiety to minimize results bias. Results: Our results showed that 75.3% of pregnant women were anxious. The mean GAD-7 score was 8.28 ± 5. Linear regression analysis showed that for every increase in the number of previous pregnancies, there was a 1.3 increase in anxiety level (*p* < 0.001). Women with no previous miscarriages were more anxious (*p* < 0.001). Surprisingly, pregnant women who were previously infected with COVID-19 were 6% less stressed. Pregnant women with comorbid-ities were more stressed (*p* < 0.001). Low income (*p* < 0.001) and age (*p* < 0.05) were the demo-graphic factors most significantly related to increased anxiety. Conclusions: The prevalence of pregnancy-related anxiety increased threefold in Saudi Arabia due to the COVID-19 pandemic. Healthcare support should be available remotely during pandemics; pregnant women (especially those with comorbidities) should be educated about the risks of infection and complications to prevent anxiety-related complications during pregnancy.

## 1. Introduction

Birth and pregnancy complications increased by 10.2% from 2019 to 2020 as a result of the 2019 coronavirus (COVID-19) pandemic [1]. The most common complications were preterm birth, preeclampsia, cesarean delivery, and perinatal death [2]. The physiological impact of anxiety is due to exposure to stress [3]. The experience of repeated stressors, such as anxiety and depression, during the COVID-19 pandemic, may have triggered physiological stress in pregnant women. 

Although COVID-19 is a respiratory disease, it leads to many health issues, such as heart disease, renal disease, and other complications [4]. The initial COVID-19 lockdown led to social isolation and a threefold increase in anxiety in the general population, reaching a prevalence of 46% [5]. Anxiety and depression were the most reported mental health issues during the pandemic, and affected mainly women, especially pregnant women, postpartum women, women who miscarried, and women who experienced domestic violence [6,7].

Mental disorders induce physiological stress which seriously affect pregnant women and their fetuses, increasing the risk of preterm delivery and low birth weight [8]. Pregnancy-induced anxiety increases the risk of postpartum depression [9]. Given increased anxiety during the pandemic, and that pregnant women are a vulnerable population, we aimed to study the physiological and psychological factors affecting pregnancy-induced anxiety disorder during the COVID-19 pandemic and to discuss our results with regard to the recent literature about pregnancy complications during the COVID-19 pandemic.

## 2. Methods

### 2.1. Study Design

We conducted a non-experimental cross-sectional study of pregnancy-related anxiety in pregnant women during the COVID-19 pandemic. The study was approved by the Institutional Review Board in February 2021.

### 2.2. Participants and Procedures

Participants consisted of 377 pregnant women living in Riyadh, Saudi Arabia, during the COVID-19 pandemic. All participants signed a consent form that described the study’s aims and noted that their participation was voluntary, that complete anonymity was ensured, and that they were allowed to withdraw from the study at any time. Data were collected by personally approaching participants in public and private clinics from March 2021 to July 2021. 

A 25-item questionnaire consisting of closed-ended questions was administered to our participants during the interview. The questions were in Arabic and English. 

Six participants were invited to pilot-test the initial draft of our questionnaire to validate it; minor changes were made based on their feedback. 

### 2.3. Measures

Closed-ended questions consisted of four sections:Study demographics: questions related to age, nationality, education level, and job status.Physiological parameters: questions related to pregnancy history, trimester, number of previous pregnancies, number of previous miscarriages, comorbidities, and body mass index.Questions related to stress (ability to control personal life and problems) and quality of life (performing daily activities and taking care of others) during the COVID-19 pandemic.A validated scale, the generalized anxiety disorder assessment (GAD-7), was used to measure participant’s anxiety levels [10].

### 2.4. GAD-7 Scale

The GAD-7 scale is a valid tool used to identify generalized anxiety disorder. The scale can also be used to screen for panic disorder, social anxiety disorder, and posttraumatic stress disorder. It is a 7-item test scored on a scale from 0 (not at all) to 3 (every day) with a high score indicating a high degree of anxiety. The sum of all items was calculated; potential scores ranged from 0 to 21 points. Cut-off scores were used to define anxiety severity as follows: 0–4 indicated no or minimal anxiety, 5–9 indicated mild anxiety, 10–14 indicated moderate anxiety, and 15 or more indicated severe anxiety.

### 2.5. Statistical Analysis

All statistical tests were performed using the Statistical Package for Social Science (SPSS (IBM Corp. Released 2017. IBM SPSS Statistics for Windows, Version 25.0. Armonk, NY, USA: IBM Corp.)) software version 25. Responses were presented as frequencies and percentages. Chi-square and Fisher’s exact tests were used to compare responses between variables in different categorical measures. Linear regression analysis was used to compare anxiety levels with other parameters. The Student *t*-test was used when appropriate; *p* values equal to or less than 0.05 were considered significant.

## 3. Results

### 3.1. Study Demographics 

A total of 377 women were interviewed. Participants’ demographic characteristics are shown in Table 1. Age and financial status were normally distributed. Of the 40% of employed pregnant women, only 6% were health care practitioners. Most of our participants (84%) were healthy with no comorbidities.

### 3.2. Physiological Parameters 

Table 2 shows participants’ physiological characteristics. The majority of participants were multigravida women in their third trimester, and 28% had experienced at least one miscarriage. Although 42% of pregnancies were not planned, 85% of women reported being pleased about their pregnancy.

### 3.3. Anxiety Level and Its Correlation with Study Demographics

According to the GAD-7 scale for generalized anxiety testing results, 75.3% of pregnant women in our study experienced different levels of anxiety (Table 3). The mean GAD-7 score was 8.28 ± 5, which was categorized as mild anxiety and was experienced by 41.4% of our participants. We correlated anxiety levels with different study variables. A detailed GAD-7 analysis is shown in Figure 1.

Education, employment, and financial status were equally distributed among participants who experienced all anxiety levels. Education and employment were not significantly correlated with anxiety level. Pregnant women with higher incomes were less anxious than participants with lower incomes (*p* < 0.001). Age was significantly correlated with anxiety level (*p* < 0.05); anxiety levels were higher in women aged 30–39 years. Anxiety levels were similar for participants with planned and not planned pregnancies.

### 3.4. Anxiety Level and Its Correlation with Physiological Parameters

Anxiety levels were lower in primigravida women compared to multigravida women. Linear regression analysis showed that for every increase in the number of previous pregnancies, there was a 1.3 increase in anxiety level (*p* < 0.001). Women with no previous miscarriages were more anxious than women with one or more miscarriages (*p* < 0.001). Surprisingly, pregnant women who were previously infected with COVID-19 were 6% less stressed than those who were not infected. Pregnant women with comorbidities (mostly asthma, diabetes, or hypertension) were significantly more stressed than those who did not report comorbidities (*p* < 0.001).

### 3.5. Anxiety Level and Its Correlation with Attitude toward the Pandemic

Although 73% of participants reported that they were stressed during the COVID-19 pandemic, the remaining 27% (who reported that they were not stressed by the COVID-19 pandemic) also showed high anxiety levels on the GAD-7 scale (84% and 80%, respectively). Regarding vaccines, 53% of pregnant women were not willing to be vaccinated while pregnant. Interestingly, all women with moderate to severe anxiety levels were willing to be vaccinated while pregnant (*p* < 0.001).

Anxiety also affected pregnant women’s daily activities (*p* < 0.001). Our results show that 51% of anxious women had difficulty performing their daily life activities such as studying, working, or taking care of other people (Figure 2). Most of our participants (41%) reported that they did not miss a doctor appointment because of the pandemic, whereas 15% reported that they missed some doctor visits because of the pandemic (*p* < 0.001).

## 4. Discussion

To the best of our knowledge, the present study is the first to examine the prevalence of anxiety in pregnant women during the COVID-19 pandemic in Saudi Arabia. In this study, psychosocial, obstetric, and pregnancy-related factors for anxiety have been investigated. We found that COVID-19 lockdowns had a significant impact on pregnant women’s well-being, which is consistent with a previous study that showed that COVID-19 pandemic restrictions affected the well-being of pregnant women [11]. 

Before the COVID-19 pandemic, the overall global prevalence of anxiety disorders was estimated to be 7.3%; during the COVID-19 outbreak, rates of anxiety in the general population increased threefold, reaching 46% [5]. This is consistent with our study, where the prevalence of antenatal anxiety before the COVID-19 pandemic in Saudi Arabia was 23%, which increased threefold, reaching an estimated 75.3% [12]. The increase in the prevalence of anxiety in our study may have been caused by fear due to the rapid spread of COVID-19 infection and increased mortality rates and/or the imposition of home quarantine and growing financial losses, all of which increased the risk of psychiatric conditions across all societies [13]. 

As pregnancy is physiologically, physically, and psychologically stressful, pregnant women experience additional fears that may increase their anxiety, such as the fear that COVID-19 might cause anomalies, growth problems for their unborn children, and preterm delivery [14]. The anxiety experienced by pregnant women in our study might not be because of fear of the pandemic; it might be associated with reduced healthy activities such as the closure of parks and gyms, as a simple healthy lifestyle reduces anxiety and depression [15]. 

Pregnant women are at higher risk of the consequences of COVID-19, which is possibly related to the altered physiology of their immune and respiratory systems [16]. In addition, women aged 35 and above are at higher risk during pregnancy [17]. Previous studies reported that one-third of pregnant women older than 30 with comorbidities were admitted to hospital due to COVID-19 infection complications. These women had preterm deliveries, stillbirths, and cesarean sections due to COVID-19 illness [18,19]. In our study, women older than 30 with comorbidities were the most anxious, which may have increased their risk of pregnancy complications. As reported in previous studies, age significantly correlated with anxiety levels; those aged 21–40 years were the most anxious during the pandemic [20,21]. This may have been because this age group is society’s workforce and was more concerned with future consequences of the pandemic, such as their jobs and business closure [22]. 

In our study, anxious women reported that they were afraid that they or a family member would become infected, which possibly prevented them from attending clinic appointments. To overcome pregnancy-related complications due to the COVID-19 pandemic, it was recommended that pregnant women not skip prenatal care appointments; this was an issue in our study [16]. Pregnant women should be aware of the risk of severe COVID-19 illness to prevent pregnancy complications. During pregnancy, the influences of a woman’s mental health and emotions on the intrauterine environment can have severe physiological implications for her and her child’s long-term health outcomes [23]. Cardiovascular and neuroendocrine reactivity to acute stress in pregnancy increases the risk of hypertension, diabetes, and postpartum depression [3]. Newborns of women with intense emotions during pregnancy showed abnormal neural development [24]. Furthermore, children of women with mental disturbances during pregnancy are at high risk for growth restriction, lower Apgar scores, and cognitive disorders [25]. In addition, children born to mothers who experienced prenatal anxiety might develop brain structure and function alterations that affect their behavior of [26]. 

Women with high-risk pregnancies (especially those with previous miscarriages) are more anxious than other pregnant women [27]. Our study found that women with no previous miscarriages were more anxious than women with a history of one or more miscarriages were; this contradicts previous reports. This might be because anxiety regarding the pandemic had more impact on women’s anxiety levels than pregnancy-related traumatic stress.

Proper screening for the mental well-being of pregnant women is crucial to improving their pregnancy, childbirth, and post-delivery experiences [28]. Virtual social and psychological support are recommended to decrease pregnant women’s anxiety during pandemics [29]. Psychosocial support provided remotely using different communication technologies improves mood and confidence in pregnant women during pandemics [11]. Identifying women at risk of depressed mood early in pregnancy is crucial to preventing childbirth and postpartum complications [30]. Therefore, it is essential to screen pregnant women for mental disturbances and provide support for pregnant women’s mental well-being; this is especially important for women with previous psychopathological disorders [31]. 

## 5. Recommendations

More research is encouraged to screen for pregnancy-related anxiety and its serious adverse effects, especially during pandemics. An ongoing multicenter longitudinal study to examine mental health and physiological stress by monitoring cardiovascular and neuroendocrine functions of pregnant women would provide a unique opportunity to compare the prevalence of mental health issues before, during, and after pandemics. A retrospective study to determine the prevalence of birth defects and pregnancy complications during pandemics is also recommended. 

## 6. Limitations

This study was limited in that it only included pregnant women in the Riyadh area. Although the GAD-7 scale is internationally validated to measure anxiety levels, it cannot provide a clinical diagnosis for anxiety disorder. As this was a cross-sectional study, a causal relationship between variables could not be measured. 

## 7. Conclusions

The prevalence of pregnancy-related anxiety increased threefold in Saudi Arabia due to the COVID-19 pandemic. Anxiety varies across pregnancies. Given the association between pregnant women’s mental health and physiological risk factors to pregnant women and their fetuses, there is an urgent need to support pregnant women to mitigate long-term adverse outcomes, especially during pandemics. Despite a decline in face-to-face psychological support, remote healthcare psychological support should be available and screening for perinatal physiological stress and mental health issues should be prioritized. Healthcare professionals should increase mental health awareness for pregnant women and emphasize the importance of attending doctors’ appointments to prevent pregnancy-related complications, especially during pandemics. 

## Figures and Tables

**Figure 1 ijerph-19-12119-f001:**
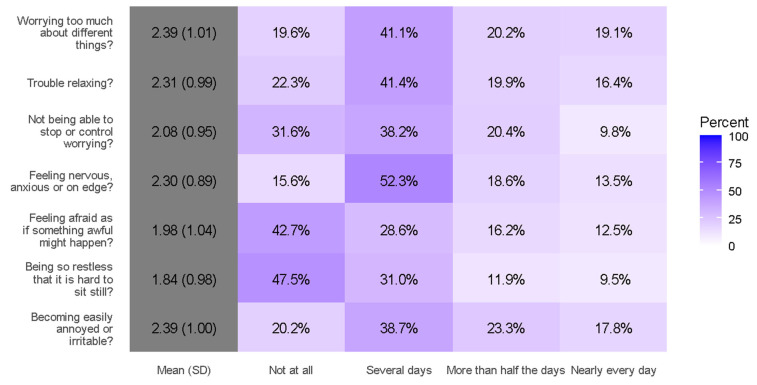
Responses to individual GAD-7 items.

**Figure 2 ijerph-19-12119-f002:**
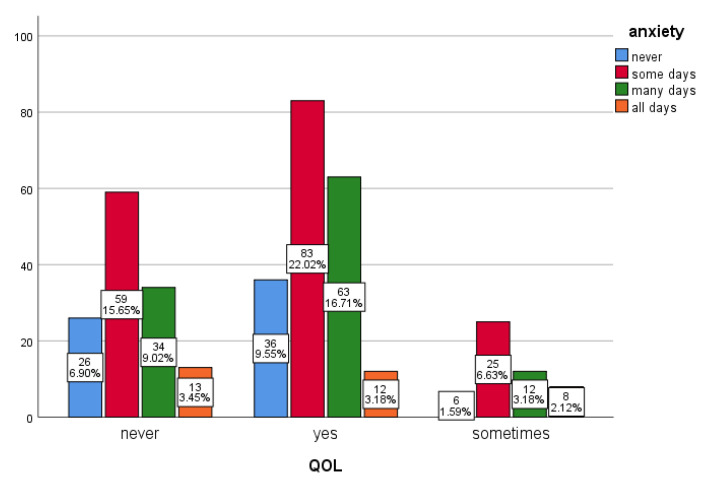
Anxiety level versus quality of life (QOL). Pregnant women whose quality of daily life and caring for others were affected during the pandemic answered “yes”, indicating higher anxiety levels than those who answered “never” or “sometimes” (*p* < 0.001).

**Table 1 ijerph-19-12119-t001:** Participants’ (*n =* 377) demographic characteristics.

Characteristic	*n* (%)
Nationality:	
Non-Saudi	17 (4.51)
Saudi	360 (95.5)
Age:	
18–26	115 (30.5)
27–34	133 (35.3)
35–43	120 (31.8)
>43	9 (2.4)
Education:	
Primary	2 (0.53)
Secondary school	6 (1.59)
High school	51 (13.5)
University degree	266 (70.6)
Post-graduate	52 (13.8)
Employment:	
Health care practitioner	21 (5.57)
Non health care practitioner	130 (34.5)
Student	40 (10.6)
Unemployed	186 (49.3)
Monthly income:	
<5000 SAR	138 (36.6)
5000–10,000 SAR	129 (34.2)
>10,000 SAR	110 (29.2)
Accommodation type:	
Owned residence	214 (56.8)
Rented housing	163 (43.2)
Living with husband:	
No	4 (1.06)
Yes	373 (98.9)
Existence of comorbidities such as diabetes and hypertension:	
No	318 (84.4)
Yes	59 (15.6)
Previously infected with COVID-19:	
Yes	100 (26.5%)
No	277 (73.5)

**Table 2 ijerph-19-12119-t002:** Pregnancy-related characteristics (*n =* 377).

Characteristics	*n* (%)
Pregnancy trimester:	
1 to 3	108 (28.6)
4 to 6	116 (30.8)
7 to 9	153 (40.6)
Pregnancy order:	
First	135 (35.8)
Second to fourth	155 (41.1)
Fifth to seventh	76 (20.2)
Eighth and more	11 (2.92)
Previous miscarriages:	
None	272 (72.1)
1 to 3	99 (26.3)
>3	6 (1.59)
Planned pregnancy:	
No	157 (41.6)
Yes	220 (58.4)
Pregnancy news had a negative impact:	
No	320 (84.9)
Yes	57 (15.1)

**Table 3 ijerph-19-12119-t003:** Participants’ degrees of anxiety.

GAD-7 Category:	*n* (%)
None (0–4)	93 (24.7)
Mild (5–9)	156 (41.4)
Moderate (10–14)	77 (20.4)
Severe (>15)	51 (13.5)

## Data Availability

Data in this research are available in tables section of this manuscript.

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
