# Peer review of "Pregnancy Complications in Pandemics: Is Pregnancy-Related Anxiety a Possible Physiological Risk Factor?"

_ijerph, 2022, doi:10.3390/ijerph191912119_

Round 1

Reviewer 1 Report

You have put a lot of work into this study. I hope that you can find a place to publish it.

377 women were recruited but what was the denominator? Were these clinic in a public or private hospital. What were the demographics of women who did not complete the questionaire?

It is best to avoid the word abortion. Use either miscarriage or termination of pregnancy depending on what you mean. I suspect you mean miscarriage. 

Pregnant women with Covid are at more risk than non pregnant women with covid but I am not sure you can state that pregnant women are increased risk of Covid. They are at increased risk of the consequences of Covid.

It is difficult to come to the conclusions you have come to based on one survey in pregnancy per woman. This was done at different gestations and ideally should have been done on all women at the same gestation (e.g. booking) or several times in the same women to have the individuals as their own controls

You state that GAD-7 tool is a valid tool to identify generalized anxiety disorder. However it is not clear that it is valid to test for covid specific related anxiety. Data comparing GAD in a cohort now that Covid has resolved may be a interesting comparison. How do you know that the GAD score relates to covid anxiety as opposed to anxiety about the pregnancy?

You have made the statement in the conclusion that pregnant women are more likely to be hospitalized with Covid. I accept this is true but is that a conclusion of your study? 

Author Response

Reviewer 1:

Comment:

It is best to avoid the word abortion. Use either miscarriage or termination of pregnancy depending on what you mean. I suspect you mean miscarriage. 

Response:

Changed as recommended by the reviewer

Comment:

Pregnant women with Covid are at more risk than non pregnant women with covid but I am not sure you can state that pregnant women are increased risk of Covid. They are at increased risk of the consequences of Covid.

Response:

Changed as recommended by the reviewer (highlighted in the discussion section)

Comment:

It is difficult to come to the conclusions you have come to based on one survey in pregnancy per woman. This was done at different gestations and ideally should have been done on all women at the same gestation (e.g. booking) or several times in the same women to have the individuals as their own controls

Response:

Changed as recommended by the reviewer (highlighted in the abstract section)

Comment:

You state that GAD-7 tool is a valid tool to identify generalized anxiety disorder. However it is not clear that it is valid to test for covid specific related anxiety. Data comparing GAD in a cohort now that Covid has resolved may be a interesting comparison. How do you know that the GAD score relates to covid anxiety as opposed to anxiety about the pregnancy?

Response:

Thank you for your comment.  We appreciate your valuable recommendation, which would be an interesting study that could be conducted in the near future.  The current study aimed to find the prevalence of anxiety during the COVID-19 pandemic, not COVID-19-related anxiety.  This scale has been used previously in previous pandemics and similar studies as ours.  The tool's validity measures generalized self-reported anxiety to estimate the prevalence and not for medical diagnosis. 

Comment:

You have made the statement in the conclusion that pregnant women are more likely to be hospitalized with Covid. I accept this is true but is that a conclusion of your study? 

Response:

statement removed as recommended by the reviewer (highlighted in the abstract section)

Comment:

You have put a lot of work into this study. I hope that you can find a place to publish it.

377 women were recruited but what was the denominator? Were these clinic in a public or private hospital. What were the demographics of women who did not complete the questionaire?

Response:

The study was done in public and private clinics.  Because the survey was filled by physically approaching participants in clinics (face-to -face), most participants approved to participate except for non-pregnant women visiting the clinic. 

Reviewer 2 Report

- Lines 29-30: The sentence is misleading because according to the cited paper, birth and pregnancy-related complications increased in 2020 compared to 2019. Here it reads as if the complication had already increased in 2019.
- Line 36: Which lockdown is being referred to? There have been multiple lockdowns around the world with varying lengths and time frames.
- Line 55: Please provide the time frame of recruitment and how and where (hospitals?) the pregnant women were recruited. Did recruitment occur during a nationwide lockdown?
- Line 68: Do you really mean abortions here or rather miscarriages or spontaneous abortions? If miscarriages are meant, please revise elsewhere in the text, as an abortion implies a voluntary termination of the pregnancy.
- There is a missing description of the QoL questionnaire.
- Is there any Covid-19 related demographic information? For example, previous infections, hospitalizations, immunization status? Since in line 115-117 the result is that previously infected women were less "stressed", it would be nice to see how many were infected.
- Sometimes the term "stressed" is used instead of "anxious". I don't think these terms are interchangeable as they describe different emotional states. Or is this reporting on the results of the QoL questionnaire rather than the GAD scale? If so, this should be mentioned and described in more detail.
- Line 135: One third of pregnant women are hospitalized. This result is not mentioned before!
- Line 138-139: "In our study, women with comorbidities and ages older than 30 are the most anxious, which might increase their risk of pregnancy complications." This does not represent the end of the story. As women age, high-risk pregnancies are very common. Women over 35 are referred to as having a "geriatric pregnancy." So maybe it's not just the pandemic, but age that makes women more anxious because age leads to more complications.
- Line 142: " most anxious pregnant women reported missing their appointment " Again, this finding is not mentioned in the results section.
- Line 158-189: In my opinion, several parts of these paragraphs would be better placed at the beginning of the introduction, as this is where the research question, the aim of the study, and a summary of the results are presented. After that, the detailed results should be placed, such as in lines 134-144. In general, the authors could try to reorganize the whole (!) discussion, because at the moment the red thread gets lost.
- Line 177: " We found that COVID-19 lockdown had a significant impact on pregnant women's well-being, especially for first-time parents " This is incongruent with the results in lines 112-114, where multiparous women are more anxious and there is an increase in anxiety with each increase in the number of previous pregnancies.
- Line 181: Here, the age group 21-40 is the most anxious, while in line 109 it is the group 30-39.
- Line 202: Is longitudinal instead of cross-sectional meant here?
- The entire manuscript is in urgent need of linguistic revision!

Author Response

Reviewer 2:

Comment:

Lines 29-30: The sentence is misleading because according to the cited paper, birth and pregnancy-related complications increased in 2020 compared to 2019. Here it reads as if the complication had already increased in 2019.
Response:

Changed as recommended by the reviewer (highlighted)

Comment:

- Line 36: Which lockdown is being referred to? There have been multiple lockdowns around the world with varying lengths and time frames.
Response:

Changed as recommended by the reviewer (highlighted). 

Comment:

- Line 55: Please provide the time frame of recruitment and how and where (hospitals?) the pregnant women were recruited. Did recruitment occur during a nationwide lockdown?
Response:

Changed as recommended by the reviewer (highlighted).  Data collection started after the second wave of COVID-19 pandemic. (the second wave was between August 2020 to February 2021).

Comment:

- Line 68: Do you really mean abortions here or rather miscarriages or spontaneous abortions? If miscarriages are meant, please revise elsewhere in the text, as an abortion implies a voluntary termination of the pregnancy.
Response:

Changes as recommended by the reviewer.

Comment:

- There is a missing description of the QoL questionnaire.
Response:

Changed as recommended by the reviewer (highlighted).

Comment:

- Is there any Covid-19 related demographic information? For example, previous infections, hospitalizations, immunization status?
Response:

Thank you for your comment.  The COVID-19 vaccine was unavailable for pregnant women during the data collection period and was obligatory only for first-line respondents. As for other diseases than COVID-19, all residents are immunized as completing immunization is obligatory to receive education in our country. We also asked the participants if they got infected, if they had any comorbidities, and if they would receive a vaccination if it were available for pregnant women.    

Comment:

Since in line 115-117 the result is that previously infected women were less "stressed", it would be nice to see how many were infected.

Response:

Added as recommended by the reviewer (highlighted in table 1)

Comment:

- Sometimes the term "stressed" is used instead of "anxious". I don't think these terms are interchangeable as they describe different emotional states. Or is this reporting on the results of the QoL questionnaire rather than the GAD scale? If so, this should be mentioned and described in more detail.
Response:

Changed as recommended by the reviewer (highlighted).  Actually, stress scale was adopted from 4-item perceived stress scale (PSS).

Comment:

- Line 135: One third of pregnant women are hospitalized. This result is not mentioned before!

Response:

Dear reviewer, the statement is rephrased (highlighted).

Comment:

- Line 138-139: "In our study, women with comorbidities and ages older than 30 are the most anxious, which might increase their risk of pregnancy complications." This does not represent the end of the story. As women age, high-risk pregnancies are very common. Women over 35 are referred to as having a "geriatric pregnancy." So maybe it's not just the pandemic, but age that makes women more anxious because age leads to more complications.
Response:

Changed as recommended by the reviewer (highlighted). 

Comment:

- Line 142: " most anxious pregnant women reported missing their appointment " Again, this finding is not mentioned in the results section.
Response:

Dear reviewer, this finding was mentioned in the results (highlighted).

Comment:

- Line 158-189: In my opinion, several parts of these paragraphs would be better placed at the beginning of the introduction, as this is where the research question, the aim of the study, and a summary of the results are presented. After that, the detailed results should be placed, such as in lines 134-144. In general, the authors could try to reorganize the whole (!) discussion, because at the moment the red thread gets lost.

Response:

Thank you for your comment.  The whole discussion was rearranged as recommended by the reviewer.

Comment:

- Line 177: " We found that COVID-19 lockdown had a significant impact on pregnant women's well-being, especially for first-time parents " This is incongruent with the results in lines 112-114, where multiparous women are more anxious and there is an increase in anxiety with each increase in the number of previous pregnancies.
Response:

Removed as recommended by the reviewer (highlighted). 

Comment:

- Line 181: Here, the age group 21-40 is the most anxious, while in line 109 it is the group 30-39.
Response:

Thank you for your comment.  Statement rephrased as recommended by the reviewer (highlighted). 

Comment:

- Line 202: Is longitudinal instead of cross-sectional meant here?
Response:

Changed as recommended by the reviewer (highlighted). 

Round 2

Reviewer 1 Report

This study is better presented now.